# Prognostic Significance of Lymphocyte Infiltrate Localization in Triple-Negative Breast Cancer

**DOI:** 10.3390/jpm12060941

**Published:** 2022-06-08

**Authors:** Toni Čeprnja, Ivana Mrklić, Melita Perić Balja, Zlatko Marušić, Valerija Blažićević, Giulio Cesare Spagnoli, Antonio Juretić, Vesna Čapkun, Ana Tečić Vuger, Eduard Vrdoljak, Snježana Tomić

**Affiliations:** 1Department of Pathology, Forensic Medicine and Cytology, University Hospital Center Split, School of Medicine, University of Split, 21000 Split, Croatia; ivana.mrklic@mefst.hr (I.M.); stomic@mefst.hr (S.T.); 2Department of Pathology, University Hospital Center “Sestre Milosrdnice”, 10000 Zagreb, Croatia; melitabalja@rocketmail.com; 3Department of Pathology, Zagreb University Hospital Center, 10000 Zagreb, Croatia; marusic_zlatko@yahoo.com; 4Department of Pathology, University Hospital Osijek, 31000 Osijek, Croatia; valerija.blazicevic@gmail.com; 5CNR Institute “Translational Pharmacology”, 00133 Rome, Italy; gcspagnoli@gmail.com; 6Department of Oncology, Clinical Hospital “Sveti Duh”, School of Medicine, University of Zagreb, 10000 Zagreb, Croatia; ajuretic@kbsd.hr; 7Department of Nuclear Medicine, University Hospital Centre Split, School of Medicine, University of Split, 21000 Split, Croatia; vesna.capkun@gmail.com; 8Department of Oncology, University Hospital Center “Sestre Milosrdnice”, 10000 Zagreb, Croatia; ana.tecic@yahoo.com; 9Department of Oncology, University Hospital Center Split, University of Split, 21000 Split, Croatia; edo.vrdoljak@gmail.com

**Keywords:** triple-negative breast cancer, cancer testis antigens, immunotherapy, NY-ESO-1, PD-L1, tumor-infiltrating lymphocytes, lymphoid aggregates

## Abstract

High infiltration by tumor-infiltrating lymphocytes (TILs) is associated with favorable prognosis in different tumor types, but the clinical significance of their spatial localization within the tumor microenvironment is debated. To address this issue, we evaluated the accumulation of intratumoral TILs (itTILs) and stromal TILs (sTILs) in samples from 97 patients with early triple-negative breast cancer (TNBC) in the center (sTIL central) and periphery (sTIL peripheral) of tumor tissues. Moreover, the presence of primary and secondary lymphoid aggregates (LAs) and the expression levels of the cancer testis antigen (CTA), NY-ESO-1, and PD-L1 were explored. High infiltration by itTILs was observed in 12/97 samples (12.3%), unrelated to age, Ki67 expression, tumor size, histologic type and grade, and LA presence. NY-ESO-1 was expressed in tumor cells in 37 samples (38%), with a trend suggesting a correlation with itTIL infiltration (*p* = 0.0531). PD-L1 expression was detected in immune cells in 47 samples (49%) and was correlated with histologic grade, sTILs, and LA formation. The presence of primary LAs was significantly correlated with better disease-free survival (DFS) (*p* = 0.027). Moreover, no tumor progression was observed during >40 months of clinical follow up in the 12 patients with high itTILs or in the 14 patients with secondary LAs. Thus, careful evaluation of lymphoid infiltrate intratumoral localization might provide important prognostic information.

## 1. Introduction

Triple-negative breast cancer (TNBC) is a subtype of breast cancer lacking hormone and HER2 receptor expression and is associated with aggressive clinical behavior and limited treatment options [1,2,3,4,5,6]. Until recently, chemotherapy was the only treatment option for these patients. Currently, however, immunotherapy based on checkpoint inhibitors is emerging as an effective new treatment modality for cancer patients and is used, among others, for the treatment of TNBC patients.

Tumor-infiltrating lymphocytes (TILs) are a vital component of the cellular anticancer immune response. In recent years, TILs have been proposed to provide prognostic value in several malignancies including melanomas and carcinomas of the upper and lower gastrointestinal tract [7,8,9]. Previous reports suggest that in early ER- breast cancers, intratumoral TIL (itTIL) presence is associated with improved disease-free survival (DFS) [10]. Moreover, CD8+ T-cell infiltration within the tumor has been reported to be correlated with reduced cancer-specific mortality in TNBC [11]. Remarkably, however, in a number of published studies, TIL infiltration in breast cancer was mostly assessed on tissue microarrays (TMAs), which fail to adequately represent tumor heterogeneity [11,12,13]. Therefore, although high TIL infiltration has been frequently associated with longer DFS and response to therapy [14,15,16,17,18], little is known about the clinical significance of their spatial localization within the tumor microenvironment and the presence of primary and secondary lymphoid aggregates (LAs).

Cancer testis antigens (CTAs) are encoded by a group of genes expressed physiologically in human germline cells and aberrantly in various malignancies. CTA expression is highly variable and may be frequently observed in melanomas and bladder, lung, ovarian, and hepatocellular carcinomas and rarely in renal, colon, gastric, and hematological malignancies [19]. Previous studies have reported a high incidence of CTA expression in TNBC, with variable reports regarding its prognostic significance [20,21,22,23,24,25]. New York esophageal squamous cell carcinoma-1 (NY-ESO-1), also known as cancer testis antigen 1B (CTAG1B), is immunogenic and reportedly induces specific B- and T-cell immunity in patients with NY-ESO-1-expressing cancers. Many clinical trials have been performed and are currently in progress to evaluate the role of CTAs as treatment targets in various cancer types [26]. Expression of CTA in TNBC could provide the opportunity for targeted immunotherapies, but further research is needed to elucidate the mechanisms of action of CTAs in breast cancer and their interaction with the immune system.

To address these issues, in this work, we analyzed TIL infiltration in different compartments of TNBC tissues in relation to the expression of PD-L1 and NY-ESO-1 CTA by using whole tumor sections, allowing for the evaluation of the central and peripheral tumor areas as well as nontumorous tissue outside tumor borders and evaluated its prognostic significance.

## 2. Materials and Methods

### 2.1. Patients

Case records of patients with breast cancer, from four clinical centers in Croatia, who underwent surgery between January 2017 and December 2018 were retrospectively reviewed. Based on pathology reports, 124 early TNBC cases were identified, and 97 patients with available tissue blocks and clinical information were included in the study. Complete follow up was available for 81 patients with a mean duration of 43.3 months.

Disease-free survival (DFS) was defined as the interval from the date of the primary surgery to the occurrence of the first locoregional recurrence or distant metastasis.

Forty-six patients (47.4%) were treated with mastectomy, forty-eight (49.5%) with quadrantectomy, and three with tumorectomy (3.1%), with subsequent axillary lymph node dissection or sentinel lymph node biopsy (SLNB) in all patients, except one for whom we had no data. All patients undergoing breast-conserving surgery received postoperative radiotherapy. None of the patients were treated with neoadjuvant therapy. Systemic adjuvant chemotherapy was administered to 90 (92.7%) patients. Of the seven patients who did not receive adjuvant chemotherapy, three (43%) refused the treatment, and four (57%) had multiple comorbidities, making them unfit for treatment.

All histologic tumor slides were independently evaluated by two pathologists (I. M. and T. Č.) and graded according to Elston and Ellis [27]. Histological types were determined according to the World Health Organization (WHO), and staging was based on TNM classification [28,29].

The study was approved by the Ethics Committee of the Clinical Hospital Centre Split and the School of Medicine, University of Split, Croatia, and was performed in accordance with the World Health Organization’s Declaration of Helsinki of 1975 as revised in 2013 [30] and the International Conference on Harmonization Guidelines on Good Clinical Practice [31]. We fully protected the patients’ anonymity.

### 2.2. Histopathology and Immunohistochemistry

Sections from fixed, paraffin-embedded cancer tissues were stained with hematoxylin/eosin, and additional immunostaining was performed to detect PD-L1 (Ventana SP142) and to NY-ESO-1 (monoclonal antibody D8.38) (Figure 1) [32]. Immunoassays were performed on a Ventana BenchMark Ultra autostainer (Roche, Tucson, AZ, USA). HER2 status was evaluated via IHC (Ventana HER2 (4B5) Antibody, Roche, Tucson, AZ, USA) and in situ hybridization (Ventana HER2 Dual ISH DNA Probe Cocktail, Roche, Tucson, AZ, USA) when needed. Tests were scored according to ASCO/CAP guidelines [33]. ER and PR were considered positive if at least 1% of the invasive tumor cell nuclei in the sample were positive [34].

To minimize the issue of tumor heterogeneity, whole tissue sections were used to determine the accumulation of stromal and intratumoral TILs (i.e., sTILs and itTILs, respectively) and the expression of PD-L1 and NY-ESO-1 by IHC. NY-ESO-1 was considered positive if a cytoplasmic and/or nuclear reaction was detectable in ≥1% tumor cells according to the median value for NY-ESO-1 expression observed in our study. PD-L1 expression was evaluated by two pathologists (S. T. and I. M.) and considered positive if discernible PD-L1 staining of any intensity was detectable in tumor-infiltrating immune cells covering ≥1% of the tumor area occupied by tumor cells, the associated intratumoral region, and contiguous peritumoral stroma [35].

The evaluation of TILs was performed by two pathologists (I. M. and T. Č.) according to recommendations by the International TILs Working Group. [36] Areas with necrosis and technical artifacts were avoided. Polymorphonuclear leukocytes were excluded from the analysis. The total accumulation of stromal TILs (sTILs) was scored as the percentage (%) of the occupied stromal component at the infiltrative margin and in the center of the tumor (sTIL total). Additionally, the accumulation of stromal TILs was evaluated separately in the center of the tumor (sTIL central) and at the periphery at the invasive front of the tumor (sTIL peripheral). If observed in direct contact with tumor cells, tumor-infiltrating lymphocytes were counted and classified as intratumoral (itTIL). Finally, the existence of lymphoid aggregates (LAs), defined as lymphatic tissue organizing into formations discernible from surrounding TILs, at the invasive tumor margin (<5 mm from invasive tumor edge) or inside the tumor was specifically evaluated. Based on the existence of germinal centers, LAs were classified as primary or secondary LAs (Figure 2).

Ki-67 expression was scored by counting 1000 tumor cells using an Olympus Image Analyzer (magnification 400×) at the hot spots and at the periphery of the invasive component, and the data are expressed as percentages of positive cells [37].

### 2.3. Statistical Analysis

Data were analyzed using SPSS Statistics 20 (IBM, Armonk, New York, United States). Statistical significance was set at *p* < 0.05, and all confidence intervals were obtained at the 95% level. The statistical significance of differences in categorical demographic data and clinical characteristics was calculated using a chi-square test and log-rank test. If a Shapiro–Wilk test indicated significant deviation from the normal distribution of all numeric variables, the median and interquartile ranges were also used. Analysis of the significance of differences in quantitative variables between 2 groups was performed using a Mann–Whitney U test. ROC analysis was used to determine the cutoff value for Ki-67 between PD-L1-positive and PD-L1-negative patients.

## 3. Results

### 3.1. TNBC Infiltration by Lymphocytes

The TNBC cohort under investigation (*n* = 97) comprised 79 (81.4%) not otherwise specified invasive carcinomas (IBC NOS) and 18 (18.6%) invasive carcinomas of special types including 7 (7.2%) metaplastic carcinomas, 4 (4.1%) invasive lobular carcinomas, 2 (2.1%) adenoid cystic carcinomas, 1 (1%) invasive papillary carcinoma, and 4 (4.1%) apocrine carcinomas. In agreement with previous studies, a majority of TNBC cases were associated with high histological grade (80.4%) and high proliferative activity as measured by Ki-67 expression (median: 56%; range: 5–98%) [1,38,39] (Table 1).

TILs were observed in most samples included in our study and were most frequently detectable in stromal compartments in peripheral and central tumor areas (Figure 3). However, in a minority of cases, variable percentages of TILs appeared to be in close contact with tumor cells and were thus defined as intratumoral TILs (itTILs) (Figure 2). In particular, in 12/97 TNBC cases (12.3%), TILs were observed in close contact with 10% or more tumor cells (Table 1). Detection of “high” itTIL percentages (≥10%) was significantly (*p* < 0.01) correlated with a higher accumulation of total stromal TILs (sTILs) and of their further classified peripheral and central subsets. On the other hand, we did not observe any significant correlation between itTIL infiltration and age (*p* = 0.153), Ki67 expression (*p* = 0.413), tumor size (*p* = 0.709), histologic type (*p* = 0.564), histologic grade (*p* = 0.166), or primary or secondary Las (Table 1).

### 3.2. NY-ESO-1 Expression

A close interaction of itTILs with tumor cells might suggest the recognition of specific neoantigens [40] or tumor-associated antigens, including cancer testis antigens (CTAs). In previous studies, we observed high expression of NY-ESO-1 CTA in TNBC [21,22,25]. In this cohort, the expression of NY-ESO-1 was not correlated with any of the studied clinicopathological parameters, including age, Ki67 expression, tumor size, histologic grade, histologic type, clinical stage, sTIL subsets, or formation of primary and secondary LAs (data not shown). However, a trend (*p* = 0.064; OR: 3.9 {1.1–13.9}; *p* = 0.039) suggesting that a correlation between NY-ESO-1 expression and itTIL infiltration was detectable in TNBC (Table 1).

### 3.3. PD-L1 Expression

The expression of “immunological checkpoint” markers has been reported to be correlated with the inhibition of antitumor immune responses [41]. To gain insight into their role in our TNBC cohort, we evaluated PD-L1 expression in tumor-associated immune cells. PD-L1 expression was significantly correlated with higher histological grade (*p* = 0.001), higher Ki-67 expression (*p* = 0.005), and formation of primary lymphoid aggregates (*p* = 0.003). Moreover, positive PD-L1 expression was significantly associated with accumulation of total sTILs and their subsets (*p* < 0.001 for all) and “high” itTILs (*p* < 0.001). However, we did not observe any significant correlation between PD-L1 expression and age (*p* = 0.509), tumor size (*p* = 0.411), histologic type (*p* = 0.092), clinical stage (*p* = 0.795), formation of secondary LAs (*p* = 0.116), or NY-ESO-1 expression (*p* = 0.511) (Table 2).

However, interestingly, PD-L1 expression in immune cells was significantly associated with itTIL infiltration (*p* < 0001) (Table 1).

### 3.4. Prognostic Significance

Univariate survival analysis revealed that the existence of primary LAs at the periphery of the tumor was significantly associated with longer DFS (LR = 4.9, *p* = 0.026; LR = 4.9, *p* = 0.027) (Table 3). Moreover, none of the 12 patients with “high” itTILs (≥10%) or the 14 patients with secondary LAs experienced distant metastasis or local recurrence within the >40 months of clinical follow up. No other examined parameter showed prognostic significance (Table 3).

## 4. Discussion

Triple-negative breast cancer (TNBC) and HER2+ breast cancer are characterized by higher levels of TILs, including T, B, and natural killer (NK) cells [42,43,44], than hormone receptor-positive cancers [45,46,47,48]. To standardize TIL evaluation in breast cancer, the International TILs Working Group has recommended a focus [36] on the accumulation of total stromal TILs, while analyses of immune cells in direct contact with the tumor (itTILs) or in the invasive tumor margin and of LAs do not currently qualify as standard assessments [36].

Indeed, stromal TIL evaluation is highly reproducible [36,49,50], whereas itTILs are difficult to identify in routine practice without additional immunohistochemical staining, are less frequently detectable, and are present in lower percentages [36,49,50]. Furthermore, since the accumulation of itTILs typically parallels that of stromal TILs, the International TILs Working Group considers that scoring itTILs does not provide important additional information [36]. However, the correlation between stromal and itTIL infiltration in breast cancer is still debated [51,52].

Since no standard thresholds for TIL evaluation in breast cancers are currently available, we analyzed them as continuous variables. In accordance with previous reports [51], we observed higher sTIL infiltration at the invasive front than in the central region of the tumors. Notably, there is no evidence that TILs at the invasive tumor edge are functionally different from those in the center, although this issue deserves further research [36].

A “high” itTIL (≥10%) infiltration was observed in 12% of tumors in our study as previously described [51] and significantly correlated with higher total or central/peripheral sTIL accumulation. Liu S. et al. examined the role of TIL in a large cohort, including breast cancers of all four major intrinsic biological subtypes, and found that besides a correlation with estrogen receptor negativity and core basal intrinsic subtype, the presence of itTILs was significantly correlated with young age and high tumor grade [10]. We did not observe any correlation between “high” itTILs and age, proliferation index, histological grade, histological type, or clinical stage. Intriguingly, all tumors with “high” itTILs showed high PD-L1 expression in immune cells. Most importantly, none of the 12 patients with itTIL ≥ 10% showed evidence of local recurrence or distant metastasis during the >40-month follow up. Although statistical validation of these findings will require analysis of larger patient cohorts, our data suggest that itTIL evaluation might be a clinically relevant prognostic factor in TNBC as previously reported by Liu et al., who found that accumulation of itTIL was associated with a significantly better outcome in women with core basal triple-negative tumors [10]. On the other hand, although Vihervuory et al. reported that a fraction of stromal TILs in the central area of the tumor was significantly associated with favorable outcome of disease, we could not confirm a significant prognostic impact of sTILs, despite a careful assessment in different stromal compartments [51].

LAs are not routinely evaluated in TIL assessment. The presence of LAs is associated with favorable outcomes in colon cancer, but its significance in TNBC has not yet been convincingly explored [53,54,55]. Notably, however, an association of LAs with a higher grade of breast cancer has previously been reported [56]. Moreover, maturation of B cells has been shown to occur intratumorally in breast tumors, leading to the production of antibodies against tumor antigens [57], possibly following the induction of T-cell immune responses [58,59].

In our study, primary LAs were detected in 68% of samples, and secondary LAs with germinal centers were detected in 14% of samples. Importantly, we observed a significant correlation between the presence of primary LAs and DFS in our cohort of patients. In addition, in 14 patients with secondary LAs, no disease progression was observed during the clinical follow up, further supporting the prognostic value of lymphoid structures in TNBC [60]. Therefore, analysis of primary and secondary LAs should not be overlooked, as their detection could have significant connotations.

PD-L1 triggering inhibits T-cell responses [61]. In agreement with previous studies [62,63,64], we observed a ≥1% PD-L1 immune cell score in 49% of samples, with a significant correlation with aggressive clinicopathologic characteristics, including higher histological grade and higher proliferative activity, as assessed by Ki-67 staining [65,66]. Moreover, PD-L1 ICS significantly correlated with sTIL accumulation at both the periphery and center of tumors, and all tumors with “high” itTILs were PD-L1 positive [66,67]. PD-L1 expression was also significantly correlated with the accumulation of primary LAs. Since it is known that PD-L1 expression can be induced by cytokines, such as IFNgamma, produced by activated T cells, PD-L1 expression might paradoxically result from ongoing antitumor immune responses.

Breast cancer cells have previously been shown by us and others to express CTAs, such as NY-ESO-1, a CTA inducing specific B-cell and T-cell immunity, which is considered a potential target for cancer immunotherapy [26,68].

In previous studies, NY-ESO-1 expression was reported in 9.3% to 28.6% of TNBC cases [21,22,25,69,70,71,72,73,74]. In this study, we observed positive NY-ESO-1 expression in 38% of samples, and we considered the expression of NY-ESO-1 positive if cytoplasmic and/or nuclear reaction was found in ≥1% of tumor cells. At present, there are no clinically relevant thresholds that could be used for the evaluation of NY-ESO-1 expression and the cutoff value used in our study was the median percentage of NY-ESO-1-positive tumor cells observed in our cohort.

Little is known about the biological functions of NY-ESO-1, although its involvement in cell cycle progression and growth has been suggested [68]. Nevertheless, the association between NY-ESO-1 expression and clinical pathological tumor features is debated. Tessari et al. reported that NY-ESO-1-positive tumors were of high grade and associated with nodal involvement [25]. We did not find a significant correlation between the expression of NY-ESO-1 and assessed clinicopathologic parameters, which is consistent with our previous report based on a different TNBC cohort and a study by Hee Jin Lee et al. [39,73].

However, interestingly, NY-ESO-1 expression was correlated with “high” itTIL infiltration. Only a few studies have investigated a correlation between NY-ESO-1 and TILs in TNBC [73]. To the best of our knowledge, the association between NY-ESO-1 expression in tumor cells and itTIL infiltration has not been described thus far. Low TIL infiltration and the absence of NY-ESO-1 expression were reported to be significantly associated with poor DFS [73]. We did not observe a significant correlation between NY-ESO-1 expression and clinical outcome, in accordance with previous studies [22]. Still, this association should be further investigated due to the hypotheses that some tumor antigens “enhance” the immune response, and the results of this study support previous opinions that NY-ESO-1 could have high levels of immunogenicity. With tumor vaccines being investigated at an increasing pace, more research is required to further our knowledge of these potential targets and their significance.

A main limitation of our study is represented by the relatively small size of our study cohort. Nevertheless, our data provide preliminary evidence of a potentially high clinical relevance of itTIL infiltration and LAs in TNBC and pave the way toward larger collaborative studies.

## 5. Conclusions

TNBC has limited therapeutic options, and innovative targeted therapies are being researched intensively. The results of our study support the prognostic relevance of itTIL in TNBC. Therefore, their analysis could become part of routine TIL assessment. Moreover, a significant correlation was observed between PD-L1 and NY-ESO-1 expression, suggesting an ongoing interaction of immune and tumor cells possibly resulting in TIL exhaustion.

The accumulation of lymphoid aggregates is a well-known positive prognostic factor in colon cancer, and it was associated with favorable outcome in our study as well. Although clinical significance could not be established in our relatively small cohort, we found that none of the patients with high itTIL accumulation or secondary LA formation had progressing disease during the 48 month median month follow-up period. Thus, while accumulation of LAs and itTILs are not reported in routine pathology examination, they might provide significant prognostic information.

## Figures and Tables

**Figure 1 jpm-12-00941-f001:**
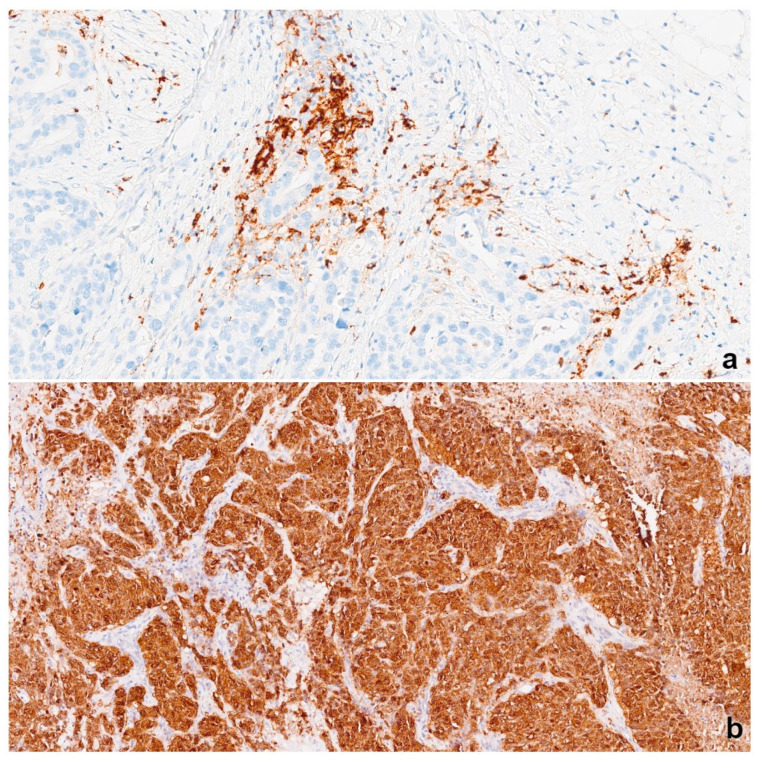
(**a**) Strong punctate and granular cytoplasmatic staining of intratumoral lymphocytes by Ventana SP142 assay. PD-L1 expression was considered positive if staining of any intensity was noted in tumor-infiltrating immune cells covering ≥1% of the tumor area occupied by tumor cells, the associated intratumoral region, and contiguous peritumoral stroma. (**b**) Strong cytoplasmic immunohistochemical staining of NY-ESO-1 in tumor cells. NY-ESO-1 expression was considered positive if ≥1% of tumor cells showed cytoplasmic and/or nuclear positivity.

**Figure 2 jpm-12-00941-f002:**
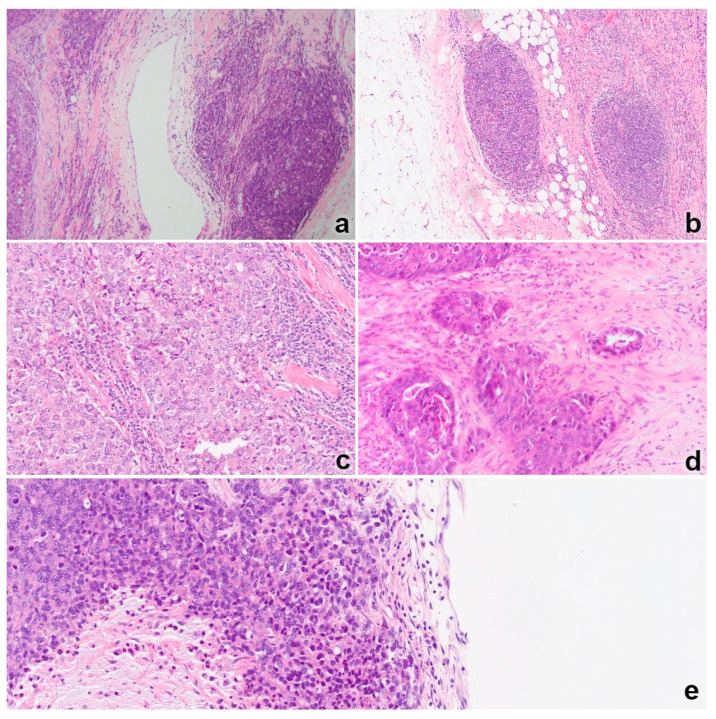
Tumor microenvironment: (**a**) secondary lymphoid aggregate with germinal center found on the invasive front of a tumor; (**b**) the primary lymphoid aggregate formation is easily discernible from the surrounding stromal TIL; (**c**) high stromal TILs occupying most of the stromal compartment on the periphery of the tumor (sTIL peripheral) and between islands of tumor cells (sTIL central); (**d**) low stromal TILs; (**e**) high intratumoral TILs defined as TIL in close contact with more than 10% of tumor cells.

**Figure 3 jpm-12-00941-f003:**
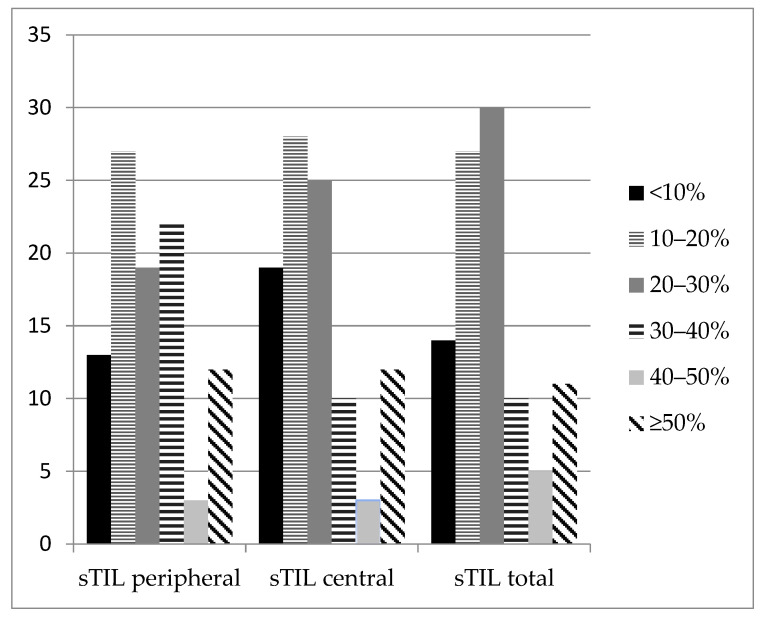
Distribution of peripheral sTILs, central sTILs, and total sTILs in TNBC (*n* = 97). The total accumulation of stromal TILs (sTILs) was scored as a percentage (%) of the occupied stromal component at the infiltrative margin and in the center of the tumor (sTIL total). Additionally, the accumulation of stromal TILs was evaluated separately in the center of the tumor (sTIL central) and at the periphery at the invasive front of the tumor (sTIL peripheral).

**Table 1 jpm-12-00941-t001:** Association of the studied variables with intratumoral TIL.

Variables		itTIL < 10% (85)	itTIL ≥ 10% (12)	*p*	OR	*p*
Age (years)	Median value (q1–q3; minimum–maximum)	65 (54–74; 29–91)	70 (60–80; 51–83)	0.153		
Ki67	Median value (q1–q3; minimum–maximum)	55% (35–75; 5–90)	62% (38–80; 27–98)	0.413		
Tumor size median value	Median value (q1–q3; minimum–maximum)	2.2 (1.5–3; 0.9–10)	1.9 (1.7–2.9; 1.1–4.5)	0.709		
Histologic grade *	2	18 (21)	0	0.166		
	3	66 (79)	12 (100)			
Histologic type	NOS	68 (80)	11 (92)	0.564		
	Other subtypes	17 (20)	1 (8)			
sTIL peripheral	Median value (q1–q3; minimum–maximum)	20 (10–30; 0–80)	40 (25–55; 10–70)	0.005		
sTIL central	Median value (q1–q3; minimum–maximum)	15 (10–25; 1–85)	45 (27–80; 10–80)	<0.001		
sTIL total	Median value (q1–q3; minimum–maximum)	20 (10–25; 1–85)	40 (26–64; 10–70)	<0.001		
Primary lymphoid aggregates	No	30 (35)	1 (8)	0.123		
	Yes	55 (65)	11 (92)			
Secondary lymphoid aggregates	No	75 (88)	8 (67)	0.121		
	Yes	10 (12)	4 (33)			
PD-L1	Negative	50 (59)	0	<0.001		
	Positive	35 (41)	12 (100)			
NY-ESO-1	0%	56 (66)	4 (33)	0.064	3.9 (1.1–13.9)	*p* = 0.039
	≥1%	29 (34)	8 (67)			

* Only one patient had a tumor of histological grade 1 and was excluded from analysis.

**Table 2 jpm-12-00941-t002:** Association of studied variables with PD-L1 expression.

Variables		PD-L1 Negative (50; 51%)	PD-L1 Positive (47; 49%)	*p*
Age (years)	Median value (q1–q3; minimum–maximum)	66 (54–78; 39–91)	65(55–72; 34–88)	0.509
Ki67	Median value (q1–q3; minimum–maximum)	42 (30–70; 5–90)	65 (50–80; 24–98)	0.005
Tumor size median value	Median value (q1–q3; minimum–maximum)	2.1 (1.6–4.5; 0.9–10)	2 (1.5–3; 0.9–5)	0.411
Histologic grade *	2	15 (31)	3 (6)	0.005
	3	34 (69)	44 (94)	
Histologic type	NOS	37 (74)	42 (89)	0.092
	Other subtypes	13 (26)	5 (11)	
Clinical stage **	I	20 (40)	20 (43.5)	0.795
	II	21 (42)	20 (43.5)	
	III	9 (18)	6 (13)	
sTIL peripheral	Median value (q1–q3; minimum–maximum)	15 (6.5–25; 1–35)	30 (25–50; 0–80)	<0.001
sTIL peripheral	≤25%	41 (82)	19 (40)	<0.001
	>25%	9 (18)	28 (60)	
sTIL central	Median value (q1–q3; minimum–maximum)	10 (5–30; 1–35)	25 (20–50; 2–85)	<0.001
sTIL central	≤20%	43 (86)	17 (36)	<0.001
	>20%	7 (14)	30 (64)	
sTIL total	Median value (q1–q3; minimum–maximum)	15 (9–20; 1–30)	30 (20–45; 5–85)	<0.001
sTIL total	≤20%	45 (90)	17 (36)	<0.001
	>20%	5 (10)	30 (64)	
itTIL	Median value (q1–q3; min–max)	1 (1–2; 0–5)	5 (1–10; 1–15)	<0.001
itTIL	≤2%	42 (84)	20 (43)	<0.001
	>2%	8 (16)	27 (57)	
Primary lymphoid aggregates	No	23 (46)	8 (17)	0.003
	Yes	27 (54)	39 (83)	
Secondary lymphoid aggregates	No	46 (92)	37 (79)	0.116
	Yes	4 (8)	10 (21)	
NY-ESO-1	0	33 (66)	27 (57)	0.511
	≥1	17 (34)	20 (43)	

* One patient had a tumor of histological grade 1 and was excluded from analysis. ** One patient did not undergo lymphadenectomy and was excluded from the analysis.

**Table 3 jpm-12-00941-t003:** Log-rank test and Cox regression univariate analysis for DFS in 97 TNBC.

Variables		Log-Rank Test				Cox Regression Univariate Analysis		
		Average DFS (Months) (SE)	95% CI	LR	*p*	RR	95% CI	*p*
Ki67 (ROC analysis)	≤55.5%	45.1 (1.4)	42–48	1.39	0.238	2.11	0.64–7	0.221
	>55.5%	40 (2)	36–44					
Histological grade *	2	41.9 (3.2)	35.7–48	0.404	0.525	0.698	0.189–2.6	0.590
	3	41.8 (1.3)	39–44					
Histological type	NOS	41.7 (1.3)	39–44	0.570	0.450	1.45	0.39–5.4	0.577
	Other subtypes	41.6 (3.6)	35–48					
Positive lymph nodes	No	45 (1.1)	43–48	6.4	0.011	4.1	1.3–12.8	0.017
	Yes	36.4 (3.2)	30–43					
sTIL peripheral by median value	≤25%	42.5 (1.7)	39–46	0.957	0.328	0.547	0.15–2	0.366
	>25%	42.7 (1.8)	39–46.3					
sTIL central by median value	≤20%	42.5 (1.7)	39–46	0.945	0.331	0.549	0.15–2	0.369
	>20%	42.6 (1.9)	39–46					
sTIL total by median value	≤20%	42.7 (1.6)	39–46	0.677	0.411	0.603	0.163–2.23	0.448
	>20%	42.4 (2)	38–46					
Primary lymphoid aggregates	No	37.7 (2.7)	32–43	4.9	0.027	0.319	1–9.9	0.051
	Yes	45.1 (1.3)	42.6–46					
PD-L1	Negative	41.4 (2)	37–45	2.97	0.085	0.351	0.095–1.3	0.117
	Positive	43.3 (1.5)	40–46					
NY-ESO-1	0%	42.4 (1.7)	39–46	0.841	0.359	0.518	0.14–1.9	0.324
	≥1%	43 (1.6)	40–46					

* One patient had a tumor of histological grade 1 and was excluded from analysis.

## Data Availability

The data presented in this study are available upon request from the corresponding author. The data are not publicly available due to the protection of personal information of our patients.

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
