# Peer review of "Prognostic Significance of Lymphocyte Infiltrate Localization in Triple-Negative Breast Cancer"

_jpm, 2022, doi:10.3390/jpm12060941_

Round 1

Reviewer 1 Report

The current manuscript finds its value in providing clinical importance of site-specific localization of TIL in TNBC. Authors emphasize the differential clinical outcome associated with TIL interaction with heterogeneous TME in TNBC. In addition, authors have explored expression of specific CTA like NY-ESO-1 and immune checkpoint marker PDL1 in TNBC tumor sections. However, the current report is very brief and provides a very limited view of an elaborate landscape of TIL interactions with heterogeneous TME and its effect on the fate of cancer treatment. Also, I found vague explanations of many terms, like, primary and secondary lymphoid aggregates, how they were categorized and studied and such.  

Overall, I find the manuscript to be accepted for a short communication. 

I wish the authors good luck.  

Reviewer 2 Report

Summary:

In this article authors have tried to evaluate the prognostic significance of the spatial tumor infiltrating lymphocytes (TILs) in Triple Negative Breast Cancer patients (TNBC) and the correlation with lymphoid aggregates (LAs), NY-ESO-1 and PD-L1 expression along with the variables used in clinical practice. Manuscript is written in a crude way. I have the following comments:

1.      Please expand any abbreviation used in the text (within brackets). For example, SLNB under “Patients” subsection.

2.      Why the authors have written the duration of months using “,” instead of “.

3.      Figure legends need to be written properly and precisely.

4.      Figure 3, why the representative symbol missing for 40-50% group?

5.      Majority of the observations in this report are against the literature published using larger cohort studies. Probably the study needs larger cohorts.

6.      No correlation was seen between itTIL infiltration; No NY-ESO-1 expression with any of the clinicopathological parameters or LAs. How do authors feel that itTIL spatial localization is of importance? How could it be associated with the TNBC pathobiology?

7.      Authors observed high expression of NY-ESO-1 in TNBC in previous study however this study shows no correlation with any of the clinicopathological variables. What is the significance of NY-ESO-1 in TNBC patients/tumors? Have the authors tried to know?

8.      Table 3, font style and the size both are completely different from rest of the text.

9.      The study lacks the consistent finding/novelty. In the previous report NY-ESO-1 was speculated in 9.3%-28.6% of TNBC cases, however in the current study they observe in > 1% of the cases.

10.  Discussion needs to be summarized based on the results in impactful way.

11.  Please keep the consistency in the font size among figure titles/legends throughout manuscript. Provide conclusive statements in discussion or conclusion wherever appropriate based on their variable observations comparing with others reported.

Round 2

Reviewer 2 Report

Authors have fulfilled my concerns. I recommend the manuscript in its current version for the publication in the Journal of Personalized Medicine.